# SARS-CoV-2, COVID-19 and Parkinson’s Disease—Many Issues Need to Be Clarified—A Critical Review

**DOI:** 10.3390/brainsci12040456

**Published:** 2022-03-28

**Authors:** Tsepo Goerttler, Eun-Hae Kwon, Michael Fleischer, Mark Stettner, Lars Tönges, Stephan Klebe

**Affiliations:** 1Department of Neurology, Essen University Hospital, 45147 Essen, Germany; tsepo.goerttler@uk-essen.de (T.G.); michael.fleischer@uk-essen.de (M.F.); mark.stettner@uk-essen.de (M.S.); 2Department of Neurology, Ruhr University Bochum, 44801 Bochum, Germany; eunhae.kwon@rub.de (E.-H.K.); lars.toenges@rub.de (L.T.)

**Keywords:** COVID-19, SARS-CoV2, Parkinson’s disease, viral parkinsonism, post-encephalitic parkinsonism

## Abstract

Neurological manifestations during the severe acute respiratory syndrome coronavirus 2 (SARS-CoV-2) pandemic are of interest, regarding acute treatment and the so-called post-COVID-19 syndrome. Parkinson’s disease (PD) is one of the most common neurodegenerative movement disorders worldwide. Hence, the influence of SARS-CoV-2 and the COVID-19 syndrome on PD patients has raised many questions and produced various publications with conflicting results. We reviewed the literature, with respect to symptoms, treatment, and whether the virus itself might cause PD during the SARS-CoV-2 pandemic in SARS-CoV-2-affected symptomatic PD patients (COVID-19 syndrome). In addition, we comment on the consequences in non-symptomatic and non-affected PD patients, as well as post-COVID syndrome and its potential linkage to PD, presenting our own data from our out-patient clinic.

## 1. Introduction

Over the last two years, the novel coronavirus severe acute respiratory syndrome coronavirus 2 (SARS-CoV-2) has swept across the globe, infecting millions of people in every country and causing a global pandemic that was last observed over one hundred years ago. SARS-CoV-2 causes a syndrome called COVID-19, which comprises many major and minor, mostly unspecific, symptoms. Systemic symptoms, such as fever, fatigue, cough, and dyspnea, are the most apparent symptoms at onset, affecting the cardiopulmonary system, and leading to interstitial pneumonia and acute respiratory distress syndrome [1]–life-threatening conditions that overwhelm the capacity of intensive care units (ICU). Therefore, treating this disorder and preventing severe manifestations is the major challenge in the fight against the pandemic. Besides these disease manifestations, other acute and subacute symptoms also arise during SARS-CoV-2 infections, including neurological symptoms and syndromes [2], such as encephalopathy [3], meningoencephalitis [4], ischemic stroke [5], acute necrotizing encephalopathy [6], and Guillain–Barré syndrome [7].

Parkinson’s disease (PD) is one of the most common neurodegenerative movement disorders worldwide, which particularly affects older people. Patients at a more advanced age or stage of PD are at an increased risk of SARS-CoV-2 infections, due to their impaired motor functions, comorbidities, frailty, and, hence, increased risk of pneumonia. The impact of the SARS-CoV-2 pandemic has raised many questions, regarding symptoms, treatment, and outcomes in PD patients. In addition, pathophysiological considerations have raised the question of whether the virus itself can cause PD.

## 2. Association between PD and SARS-CoV-2: Pathophysiological Considerations

One of the most specific symptoms that occurs after SARS-CoV-2 infection is altered smell (hyposmia or dysosmia) and taste (dysgeusia), affecting about 65% of all infected people [8]. Hyposmia is also the most common symptom in presymptomatic PD patients, affecting numerous patients long before they develop motor symptoms [9,10]. This has led to the assumption that there may be a linkage between these two diseases, particularly because neurodegeneration in PD is thought to occur after a pathogenic encounter, e.g., a neurotropic virus, via a nasal or gastric pathway [11]. An overlap can be drawn with the recently published case–control PET study, with the differentiation of PD into the body-first and brain-first subtypes [12]. The authors stated that in the body-first subtype, the pathogen enters the brain via the gut, and the alpha-synuclein pathology spreads from there into the brain. In the brain-first subtype, the pathogen most likely enters the brain via the olfactory bulb, and the alpha-synuclein pathology spreads from there downwards into the body. Entering the brain via non-myelinated nerves, the pathogen most likely triggers chronic neuroinflammation and cytokine production, thus activating microglia [13] (Figure 1). Neuroinflammation can be observed in autopsy studies in PD animal models [14] and, functionally, in positron emission tomography (PET) imaging in PD patients [15]. A study from 1992 demonstrated that coronavirus was present in the cerebrospinal fluid (CSF) of 18 PD patients compared to normal controls and patients with other neurological disorders [16]. SARS-CoV-2 enters and infects cells using the angiotensin-converting enzyme 2 (ACE2) receptor [17]. ACE2 receptors are expressed in neurons and glial cells in many brain regions, as well as in the substantia nigra and the olfactory bulb [18]. A preprint study showed that SARS-CoV-2 RNA could be found in brain tissue in 10 out of 11 performed autopsies, which provides evidence of substantial viral burden [19]. The invading neurons in the olfactory bulb might explain the occurrence of hyposmia, which, unlike in PD, is reversible in COVID-19. Studies showed recurrence of smelling in all patients (n = 46) who reported hyposmia 17 days after the onset of symptoms [20].

In 1917, von Economo described a syndrome called encephalitis lethargica [21,22], which was followed by the so-called ‘Spanish flu’ pandemic in 1918/1919 (caused by the H1N1 influenza virus and occurred in a significantly higher proportion of younger people than older people) [13,23]. Encephalitis lethargica included lethargy, cognitive decline, headache, and sickness; during disease progression, PD-like symptoms, such as bradykinesia, rigor, and the inability to move, also occurred [22,24]. In the 1920s, parkinsonism, due to encephalitis lethargica, was responsible for over 50% of all cases, affecting mostly young people in their third decade. Compared to idiopathic PD, this syndrome showed a coarse tremor, instead of a fine and rhythmic tremor, and more often presented with ocular movement disorders as oculogyric crises [24]. Encephalitis lethargica occurred less frequently during the following decades. Post-encephalitic parkinsonism (PEP) has been described in a few case series for several viruses, including the herpes simplex virus, cytomegalovirus, varicella zoster virus, influenza virus type A, coxsackie virus, poliovirus, human immunodeficiency virus, West Nile virus, and the Japanese encephalitis B virus [25,26]. In patients who developed PEP, an improvement in motor symptoms was observed after levodopa treatment, in the few cases administered this treatment [25,27,28,29]. Although the improvement in motor symptoms was good at the beginning, these patients also developed dyskinesia after prolonged treatment [30,31]. In contrast to PD, the symptoms associated with acute post-encephalic parkinsonism may also disappear in several cases [27,32,33]; in autopsy studies, no Lewy body formations have been found [34].

The pathogenesis of PD has been extensively discussed; in addition to genetics, neuroinflammation seems to play a likely role, whether idiopathic or triggered by a pathogen. The interactions between different causes have formed the ‘multiple hit’ theory, which assumes that many incidences, such as genetic disposition and several infections (gastric or respiratory), during a lifetime accumulate in the development of PD [11,35].

We have elaborated on these points in the following sections, and discussed important aspects of the SARS-CoV-2 pandemic and PD.

## 3. PD Patients Infected with SARS-CoV-2 and Suffering from COVID-19

Many COVID-19 studies published to date are from regions in Italy that were severely affected during the first wave of the pandemic. A higher prevalence of COVID-19 in PD patients was confirmed in a case–control survey of 740 PD patients in Tuscany [36]. Overall, 7% of the entire PD population and up to 11% of those aged 65–85 years were affected by COVID-19, compared to <1% of the Italian population in general [36]. However, a case–control study in 1486 PD patients and 1207 family members from the Lombardy region demonstrated no difference in COVID-19 rates (7.1% vs. 7.6%, respectively) [37]. In terms of severe disease after SARS-CoV-2 infection, cardiovascular comorbidities appear to be a risk factor, rather than PD itself [36,38,39]. Data from 696 PD patients, 184 with parkinsonism symptoms and 8590 age-matched controls, from Northern Italy indicated that hospitalization rates due to SARS-CoV-2 infections were similar between the PD patients (0.7%) and the controls (0.6%) [40]. However, there was a higher hospitalization rate (3.3%) in parkinsonism patients (e.g., those with progressive supranuclear palsy, multiple system atrophy or vascular parkinsonism), due to a more severe health status, greater frailty, and a higher need for care in nursing homes with higher infection rates [40]. Similarly, a small study from Spain showed that institutionalized PD patients were at a higher risk of SARS-CoV-2 infection, development of COVID-19, and suffered more often from dementia as a comorbidity [41].

In a cross-sectional study of 102 SARS-CoV-2 PCR-positive patients, we investigated the frequency, type, severity, and risk of neurological involvement in COVID-19 patients [42]. Across the cohort, 59.8% of the patients had neurological involvement. We found that 25 (24.5%) patients suffered from unspecific neurological involvement, such as worsening of a pre-existing neurological condition (8.5%, *n* = 5), including one patient with exacerbation of pre-existing PD. We additionally investigated the relationship between pre-existing neurological comorbidities and the presence or severity of neurological involvement, and found increasing frequency and severity among those patients with pre-existing neurological comorbidities, including PD. This is not surprising, as PD patients with infections show an increase in severity of the disease symptoms—an observation that is also true for PD patients suffering from COVID-19 [39,43]. Several different hypotheses, directly or indirectly associated with PD, have been proposed for this phenomenon [44]. Increasing OFF times and motor fluctuations have been reported in a community-based study with 141 PD patients in the Lombardy region of Italy, of which 12 (8.5%) patients also had COVID-19 [39]. It was assumed that the deterioration of PD symptoms observed among the COVID-19 PD patients was due either to a direct influence of the infection or impaired pharmacokinetics of the dopaminergic treatment (e.g., diarrhea) [39]. As a direct consequence, it has been suggested that dopaminergic medication should be increased or adjusted during infection [39,45]. This study only observed patients for a period of three months, so data on long-term effects, worsening, or remission were not collected. Deterioration of PD motor symptoms may also precede a diagnosis of COVID-19, as described in two PD patients with poor outcomes after subthalamic nucleus deep brain stimulation [46]. On the other hand, common symptoms of COVID-19, such as anosmia or fatigue, belong to the spectrum of non-motor PD signs. In most published studies, the mortality rate due to COVID-19 does not differ between PD patients and the general population [36,37]. However, mortality is much higher (up to 40%) in the advanced stages of PD, with advanced age and comorbidities [45,47].

## 4. Newly Diagnosed PD Patients Linked to COVID-19

Parkinsonian symptoms due to the development of COVID-19 are rare, and evidence linking COVID-19 to developing PD is currently hypothetical. We screened the literature for cases describing the new onset of PD associated with COVID-19 and identified six cases of patients with new parkinsonism motor symptoms; in five of these cases, there was evidence of impaired dopaminergic uptake in the basal ganglia after nuclear imaging (Table 1) [48,49,50,51,52,53].

These six cases described the acute onset of parkinsonian symptoms with rigidity, bradykinesia, and tremor after SARS-CoV-2 infection. The ages ranged from 35 to 72 years. In three cases [48,50,51], COVID-19 symptoms were mild to moderate; one patient had a short hospitalization of three days, while the other two patients did not need hospital treatment. In the other three cases [49,52,53], treatment in an ICU, with mechanical ventilation or oxygen therapy, was required. PD symptoms began from 5 days up to 10 weeks after the positive SARS-CoV-2 test. Four cases were treated with dopaminergic medication, which improved the symptoms in two cases [48,50]. In one case, constipation for ten years was stated as a putative prodromal symptom [51]. Follow-up data were available in three cases; one patient had no symptoms after two months of follow-up [52], another had minor symptoms that were still present after 4 months of treatment with pramipexole [48], while the third patient was still impaired and unable to walk without aid [53].

Unfortunately, there are no follow-up data for evidence of a parainfectious or chronic impairment. In two cases, impairment was reported after nine months. The latter case [53] was still unable to walk without aid after nine months. This might be due to general weakness after COVID-19 and encephalopathy, rather than to parkinsonian symptoms. In the other case, their parkinsonian symptoms improved after treatment with pramipexole, with the addition of biperiden to manage the residual resting tremor [48].

Nuclear imaging in five of these cases (using FP-CIT-SPECT and F-DOPA-PET) revealed evidence that the basal ganglia were affected, where chronic impairment of dopaminergic transmission, resulting from nigrostriatal degeneration, contributed to motor symptoms [54,55]. In one case [53], FDG-PET imaging showed diffuse cortical hypometabolism, with significant hypometabolism in the precuneus, most likely associated with dementia due to Alzheimer’s disease [56]. Congruently, this patient demonstrated cognitive decline during the progression of COVID-19, but also showed relative hypermetabolism in the basal ganglia, as often observed in PD patients [56]. Patients with acute encephalitis also show hypermetabolism in the affected brain regions and hypometabolism after remission [57,58]. Therefore, in this case [53], hypermetabolism in the basal ganglia was most likely due to the encephalopathic process. In two cases, good responses to dopaminergic medication indicated that the impairment was responsive to dopaminergic substitution, which is congruent with the nuclear imaging.

It remains unclear whether COVID-19 induced PD, whether the disease revealed undiagnosed PD, or whether the disease induced encephalitis of the basal ganglia (post-encephalitic parkinsonism). Long-term data for evidence that SARS-CoV-2 induces PD do not yet exist. PD is defined by its motor symptoms of rigidity, bradykinesia, and tremor, and its loss of dopaminergic cells in the substantia nigra (because of the accumulation of alpha-synuclein, which leads to neurodegeneration). In addition, PD progressively worsens over time, which has not been observed in these cases so far.

Clinical follow-up data to implicate a progressive neurodegenerative disease are missing in all six cases. However, at least one case had complete remission of PD symptoms [52], suggesting that an acute reversible parkinsonian condition is possible. On the other hand, nuclear imaging showed impaired dopaminergic transmission in all the examined patients, providing strong evidence for a neurodegenerative disease. Furthermore, in one case, a prodromal symptom (constipation) was present [51], indicating that undiagnosed PD must be considered, although constipation is a non-specific prodromal symptom that can have other causes.

Given the small number of reported cases and the large number of infected people worldwide, parkinsonian symptoms linked to a SARS-CoV-2 infection or COVID-19 are likely to be a very rare condition. In addition, compared to the small number of cases of viral parkinsonism caused by other viruses, post-encephalitic parkinsonism may be a neuroinflammatory reaction to the infection, rather than a virus-specific disease. However, coronaviruses have been found in the CSF; thus, infection with a virus such as SARS-CoV-2 might be linked to the onset of PD. Furthermore, only a relatively short period of time has passed since the start of the pandemic; it is possible that, in the long term, symptoms of PD may occur months after infection with SARS-CoV-2.

## 5. Impact of SARS-CoV-2 Pandemic on PD Patients with Asymptomatic SARS-CoV-2 Infection or Unaffected PD Patients

The pandemic is not only a burden for PD patients with COVID-19. PD patients infected with SARS-CoV-2, but without COVID-19, and non-infected PD patients also suffer from the current situation. PD patients are vulnerable to new life events and changes, especially negative ones. In particular, negative emotional stress can lead to an increase in symptoms, effects that can be partially explained by the impaired dopaminergic function in PD patients [59,60]. Additionally, social isolation during lockdown periods has reduced the access of PD patients to their physicians and co-therapists (physiotherapists, occupational therapists, and speech therapists). For example, during the lockdown period in Italy, many PD patients reported worsening of their motor, non-motor, and neuropsychiatric symptoms [61]. A French study analyzed the consequences of a sudden lockdown in 2653 PD patients [62]. In 88% of all the patients, physiotherapy was interrupted, and many motor symptoms and pain worsened during the observation period of four weeks between March and April 2020. Around 45% of the patients also reported that their psychiatric state (i.e., depression and anxiety) had been exacerbated by the situation.

Adaptive digital (i.e., remote) treatment strategies during the pandemic are, therefore, important, including telemedicine treatment from physicians and services provided by co-therapists. Head-to-head studies must also evaluate whether this is a long-term alternative to personal contact. The pandemic provides an opportunity to assess these treatments and the potential of wearable sensors, for example, to monitor treatment or disease progression.

## 6. Post-COVID-19 Syndrome in PD Patients

The post-COVID-19 syndrome has been defined as the “signs and symptoms that develop during or after an infection consistent with COVID-19, continue for more than 12 weeks and are not explained by an alternative diagnosis” [63]. In a small series of 23 PD patients, 85% developed post-COVID-19 symptoms, including worsening of motor functions, fatigue, and cognitive disturbances [64]. This has been the only published study addressing this condition so far. However, these results must be interpreted with caution, due to the study’s small sample size and the diagnostic criteria that are yet to be defined for post-COVID-19 syndrome in PD patients. In addition, other effects, such as isolation due to lockdown periods and reduced access to physiotherapists, must be considered.

In another unpublished study, we included 171 (mean age 43.2 ± 12.7 years (range 18–71 years)) patients who fulfilled the WHO case definition of the post-COVID-19 condition, as defined by the Delphi consensus [65]. In a subanalysis we performed on this dataset, we discovered that none of the patients described motor symptoms, such as bradykinesia, impaired posture, or tremor, after infection with SARS-CoV-2. These findings were underlined by the fact that, after diagnostic workup, PD was not diagnosed among these patients. In addition, none of the 171 post-COVID-19 patients presented previously diagnosed PD (own data).

On the one hand, this finding could be due to the younger age of the patients who presented with post-COVID-19 syndrome in our out-patient clinic, where neurodegenerative diseases do not have a high prevalence. However, we suppose that PD patients might not explicitly attribute the worsening of non-motor PD symptoms to COVID-19. Hence, we speculate that post-COVID syndrome does not play a major role in PD patients.

Taking together the differentiation between post-COVID-19 syndrome in PD and the general worsening of PD symptoms due to COVID-19 bears difficulties. Most of the post-COVID-19 syndrome-associated symptoms, such as fatigue and cognitive disturbances, overlap with the non-motor symptoms of PD. Worsening of PD symptoms in PD patients often occurs during systemic inflammation, due to a viral or bacterial infection [66,67]. Patients do not always fully recover, and may remain in a worse condition than before. In a case study with 80 PD patients with systemic inflammation, 26 patients suffered from motor deterioration, 19 of whom still suffered a persistent decline in motor functions six months after infection [66]. The following pathophysiological considerations address this issue [44]: (1) Altered dopaminergic medication, due to swallowing difficulties, reduced intestinal uptake, e.g., diarrhea, or altered the permeability of the blood–brain barrier. (2) Altered cerebral dopamine metabolism, due to altered levels of cytokines. (3) Enhanced neurodegeneration, due to enhanced neuroinflammation. Chronic neuroinflammation is a driving factor in neurodegenerative diseases [15,68], so this aspect best explains the persistent decline in PD symptoms after infection.

Similarly, the mechanism of chronic neuroinflammation is discussed in post-COVID-19 syndrome [69,70], yet studies investigating this issue are still missing.

Cases of patients who have developed parkinsonism after COVID-19, as a post-acute sequelae of COVID-19 (PASC), similar to encephalitis lethargica in the last century, have not been described yet, except in one case, who developed parkinsonism 10 weeks after SARS-CoV2-infection and had motor symptoms months later [53].

Since PD is a neurodegenerative disease, with a long latency until the onset of the first motor symptoms, it is important to ask patients with post-COVID-19 syndrome about motor symptoms.

At this stage of the pandemic, predictions are difficult to make.

## 7. Conclusions

The SARS-CoV-2 pandemic has a significant impact on PD patients, regardless of whether they have been infected by the virus. PD per se does not represent a risk factor for increased mortality associated with COVID-19, but advanced age and disease progression in a PD patient do have a negative impact. Isolation and limited access to medical care services can lead to aggravation of the symptoms in PD patients. Parkinsonian symptoms occurring in response to COVID-19 is a rare condition, and evidence linking it to the development of PD is currently hypothetical.

As a limitation, we have to state that many studies have been performed and data have been collected about PD patients and COVID-19 in the first six months of the pandemic, with cohorts of up to 2500 patients, but only a few were conducted in the following months and years. In the meantime, with millions of infected people worldwide, the number of infected PD patients is also expected to be higher. This gap must be filled, in order to understand the possible link between PD and COVID-19.

A second limitation is that most studies on PD and SARS-CoV-2 generally examined patients with idiopathic Parkinson’s disease, but no reliable data about atypical Parkinson’s disease were found.

According to our own data, Parkinson symptoms do not frequently occur in patients with post-COVID syndrome. The already broadly discussed threat of a post-pandemic Parkinson pandemic, as was the case almost a hundred years ago, cannot be confirmed at the current stage, but must be re-evaluated in the future.

## Figures and Tables

**Figure 1 brainsci-12-00456-f001:**
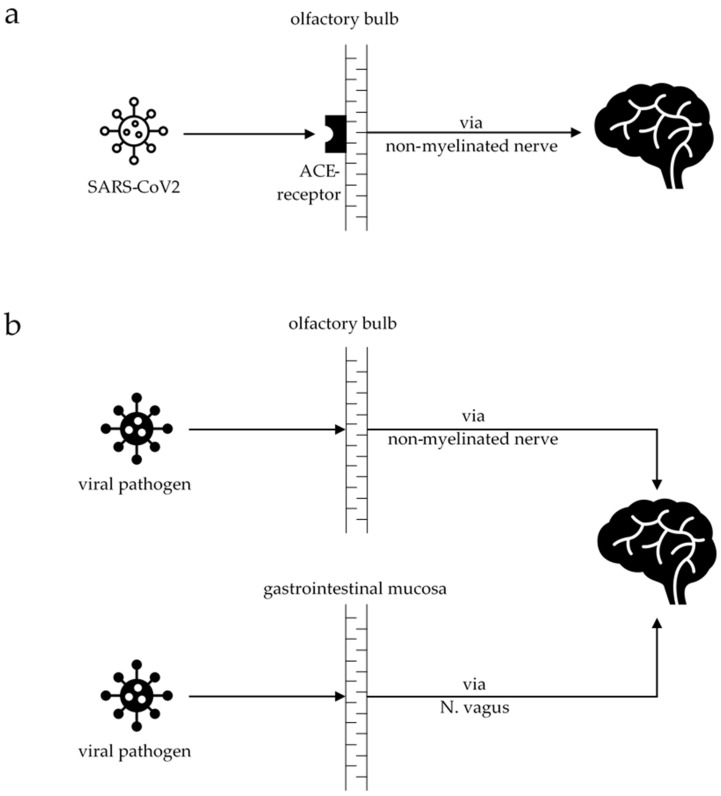
Schematic diagram of the potentially harmful pathway in COVID-19 and Parkinson’s disease (PD): (**a**) the SARS-CoV-2 virus enters the brain via the olfactory bulb, binding to the ACE receptors. From there, it spreads via non-myelinated nerves into the brain. (**b**) In PD, there are two hypothetical, not yet fully understood, routes that the viral pathogen can use to enter the brain, either via the olfactory bulb (“brain-first” theory) or via the gastrointestinal mucosa (“body-first” theory), spreading into the brain via the vagal nerve.

**Table 1 brainsci-12-00456-t001:** Published cases of COVID-19 cases related to parkinsonism.

Publication	Age	Sex	COVID-19 Severity	Length of Hospitalization	Time Until Onset of Parkinson Symptoms	Parkinson Symptoms	Response to Dopaminergic Medication	Dopaminergic Nuclear Imaging	Prodromal Symptoms/ Neuroleptic Medication	Follow Up
Cohen et al. [48]	45	male	moderate	3 days of hospitalization	ca. 3 weeks	micrographia, hypophonia, hypomimia, tremor, bradykinesia, rigidity	good; (pramipexole)	F-FOPA: decreased uptake in both putamina and the left caudate	none	rest tremor 4 months after the onset of symptoms
Mendéz-Guerrero et al. [49]	58	male	severe; mechanic ventilation required	>47 days, including time in ICU	32 days	hypomimia, bradykinesia, rest and postural tremor	poor; (apomorphine)	FP-CIT-SPECT: decreased uptake in both putamina	none	no follow-up data
Faber et al. [50]	35	female	mild	no hospitalization required	10 days	hypophonia, hypomimia, hypometric saccades, bradykinesia, gait impairment	good; (levodopa/benserazid)	FP-CIT-SPECT: decreased uptake in left putamen	none	no follow-up data
Makhoul et al. [51]	64	female	mild	no hospitalization required	5 days	hypomimia, bradykinesia, rest tremor	not given	FP-CIT-SPECT: decreased uptake in right putamen	constipation	no follow-up data
Akilli et al. [52]	72	male	severe; non-invasive ventilation required	ca. 4 weeks	5 days	rigidity, bradykinesia, rest tremor, gait impairment	not given	none	none	no symptoms 2 months after SARS-CoV2 infection
Morassi et al. [53]	70	female	moderate to severe; receiving oxygen therapy	ca. 3 weeks	ca. 10 weeks	rigidity, bradykinesia, hypomimia, hypophonia, ophthalmoparesis	modest; (levodopa/carbidopa)	FP-CIT-SPECT: decreased uptake in both putamina, more severe left side	none	9 months, still symptoms and inability to walk without aid

## Data Availability

The study is registered in the German register for clinical trials (DRKS) under the registration number: DRKS00023312.

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
