# Peer review of "SARS-CoV-2, COVID-19 and Parkinson’s Disease—Many Issues Need to Be Clarified—A Critical Review"

_brainsci, 2022, doi:10.3390/brainsci12040456_

Round 1

Reviewer 1 Report

The authors comprehensively relate to possible existing links between SARS-CoV2 infection, COVID-19 and Parkinson's Disease, which is an interesting topic in the light of already known post-COVID neurological deterioration observed in a number of individuals.

The paper itself is well structured, and consecutive paragraphs provide insight into different aspects of the pandemic and Parkinson's Disease, including the possibility of de novo PD occurrence, exacerbation of symptoms, and influence of social/medical isolation on PD patients. 

One of the strengths of the paper is that Authors' relate to their own clinical experience (Fleischer, M., et al., Observational cohort study of neurological involvement among patients with SARS-CoV-2 infection. Ther Adv Neurol Disord, 2021). However introducing unpublished, unverified data in Section 6 (Post-COVID-19 syndrome in PD patients) may be a significant source of scientific bias. To reviewer's opinion such data should be removed from the manuscript.  

Author Response

The authors comprehensively relate to possible existing links between SARS-CoV2 infection, COVID-19 and Parkinson's Disease, which is an interesting topic in the light of already known post-COVID neurological deterioration observed in a number of individuals.

The paper itself is well structured, and consecutive paragraphs provide insight into different aspects of the pandemic and Parkinson's Disease, including the possibility of de novo PD occurrence, exacerbation of symptoms, and influence of social/medical isolation on PD patients. 

One of the strengths of the paper is that Authors' relate to their own clinical experience (Fleischer, M., et al., Observational cohort study of neurological involvement among patients with SARS-CoV-2 infection. Ther Adv Neurol Disord, 2021).

  1. However introducing unpublished, unverified data in Section 6 (Post-COVID-19 syndrome in PD patients) may be a significant source of scientific bias. To reviewer's opinion such data should be removed from the manuscript.

Response to 1: Dear Reviewer, thanks for reviewing our manuscript and the positive feedback. Regarding your concerns about mentioning unpublished data.
We understand your concerns, that is why we mention specifically that the data is yet unpublished.
For this review it was important to us to state that none of the 171 patients with post-COVID-19 we saw in our outpatient clinic had Parkinson’s disease before or developed motor symptoms after COVID-19, as we found out in a subanalysis of the patients’ data. We also discussed critically the possible reasons for this. Due to this clear results we think it is, under this certain preconditions, acceptable to highlight it.
We added a sentence to clarify that we performed a subanalysis focussing on motor symptoms only and abstained from mentioning other symptoms.

Reviewer 2 Report

This is an interesting study providing a perspective on the effect of COVID and pandemic on patients affected by Parkinson disease. The paper describes carefully the evidence from literature and propose an interesting viewpoint regarding pandemic, but I have some concerns:

-the comparison of COVID-19 with encephalitis lethargica is appropriate. However, a focus on psychosis in PD is lacking. Is there evidence of psychiatric symptoms in the cohort described?

-“This is not surprising as PD patients with infections show an increase in severity of the disease symptoms – an observation that is also true for PD patients suffering from COVID-19”. I suggest to cite and discuss literature on emergencies in parkinsonism and akinetic crisis.

-“direct influence of the infection or impaired pharmacokinetics of the dopaminergic treatment”. This sentence needs to be supported by appropriate reference.

-“social isolation during lockdown periods has reduced the access of PD patients to their physicians and co-therapists (physiotherapists, occupational therapists, speech therapists).” There are several studies exploring the role of lockdown in patients affected by neurological disease (i.e. neuromuscular disorders, migraine) and their effects on nutrition, sleep and physical activity. I suggest to discuss and cite them when appropriate.

-I think that functional disorders might have a role in long COVID and post COVID syndromes. Moreover, functional Motor symptoms and psychiatric disorders are more frequent in PD patients than in general population. I suggest to discuss these data on the light of recent data from the Italian registry (“Clinical Correlates of Functional Motor Disorders: An Italian Multicenter Study. Mov Disord Clin Pract. 2020”).

-it seems that the authors discuss about patients with akinetic-rigid PD. I suggest to clearly specify, because the conclusion might not be extended for all patients with PD and fluctuations (motor and non-motor). For example, the risk of akinetic crisis is not the same in akinetic PD versus PD associated with tremor.

-the study sample for data on PD patients infected is quite low. I suggest to discuss it as a limitation. In two years of pandemic, many PD patients have been infected by COVID-19. Indeed, PD is not a rare disease and I think that more patients are required to confirm the reported findings.

-It would be interesting to report data on patients with atypical parkinsonism (MSA, PSP, CBD). These patients usually display akinetic parkinsonism and might have similar findings compared to PD.

-Grammar and style: they are overall satisfying, but I suggest to replace “can cause” with “might” in the abstract.

Author Response

This is an interesting study providing a perspective on the effect of COVID and pandemic on patients affected by Parkinson disease. The paper describes carefully the evidence from literature and propose an interesting viewpoint regarding pandemic, but I have some concerns:

Dear reviewer, thanks for revising our manuscript. Sometimes it is not clear to which paragraph your concerns apply to exactly, maybe because of layout reasons, but we tried our best to refer to each point appropriately.

  1. the comparison of COVID-19 with encephalitis lethargica is appropriate. However, a focus on psychosis in PD is lacking. Is there evidence of psychiatric symptoms in the cohort described?

Response to 1: I guess you mean the post-COVID-19 cohort of 171 patients. Yes, there is evidence of psychiatric symptoms in many of these patients, which is plausible, because psychiatric symptoms, especially cognitive impairment and fatigue, belong to the diagnostic criteria according to the WHO case definition of post-COVID-19 condition, defined by Delphi consensus (Soriano et al., 2021). Talking about yet unpublished data it is very difficult to differentiate between the ample reasons causing psychiatric symptoms in this group. That is why we looked in our subanalysis on motor symptoms only to detect possible patients with parkinsonism.
As the current stage predictions, whether some of the patients with psychiatric symptoms might develop Parkinson’s disease in the future, are very difficult to make.

  1. “This is not surprising as PD patients with infections show an increase in severity of the disease symptoms – an observation that is also true for PD patients suffering from COVID-19”. I suggest to cite and discuss literature on emergencies in parkinsonism and akinetic crisis.

Response to 2: This is indeed a very important aspect. As far as we understand your concern, we think we addressed this aspect appropriately at another passage of our manuscript (Point 6. 4th paragraph on page 7).  

  1. “direct influence of the infection or impaired pharmacokinetics of the dopaminergic treatment”. This sentence needs to be supported by appropriate reference.

Response to 3: We added the study cited (Cilia et al., 2020). 

  1. “social isolation during lockdown periods has reduced the access of PD patients to their physicians and co-therapists (physiotherapists, occupational therapists, speech therapists).” There are several studies exploring the role of lockdown in patients affected by neurological disease (i.e. neuromuscular disorders, migraine) and their effects on nutrition, sleep and physical activity. I suggest to discuss and cite them when appropriate.

Response to 4: This is an interesting point. We already cited Schirinzi et al. concerning this aspect. In general we want to keep focussed on PD itself and less with other neurological disorders, so we will not extend this paragraph on other neurological disorders. 

  1. I think that functional disorders might have a role in long COVID and post COVID syndromes. Moreover, functional Motor symptoms and psychiatric disorders are more frequent in PD patients than in general population. I suggest to discuss these data on the light of recent data from the Italian registry (“Clinical Correlates of Functional Motor Disorders: An Italian Multicenter Study. Mov Disord Clin Pract. 2020”).

Response to 6: Yes, functional disorders in general might have a role in post-COVID and the study you recommend is very interesting.
But the studies made so far and also our own data show, that PD patients are not at a higher risk developing post-COVID syndrome and healthy people do not develop motor symptoms or functional motor symptoms when having post-COVID syndrome.
We cannot support your hypothesis with appropriate data regarding motor symptoms. Focussing on PD and COVID-19 we will not discuss this.
But your suggestion will probably be discussed in our yet unpublished manuscript regarding our post-COVID study with 171 patients. 

  1. it seems that the authors discuss about patients with akinetic-rigid PD. I suggest to clearly specify, because the conclusion might not be extended for all patients with PD and fluctuations (motor and non-motor). For example, the risk of akinetic crisis is not the same in akinetic PD versus PD associated with tremor.

Response to 6: Revising all the studies we cited, we have to admit that neither one differentiated between akinetic-rigid PD or PD predominantly with tremor. The big cohort studies in Italy, France and Spain included all patients with diagnosed PD. Also the post-COVID-study by Leta et al. did not differentiate.

Therefore we cannot specify neither.

We agree with you, that this might be a huge difference concerning the risk of morbidity or mortality. 

  1. the study sample for data on PD patients infected is quite low. I suggest to discuss it as a limitation. In two years of pandemic, many PD patients have been infected by COVID-19. Indeed, PD is not a rare disease and I think that more patients are required to confirm the reported findings.

Response to 7: Yes, this is true. We added a short paragraph in our conclusion to address your concern. “As limitation we have to state is, that many studies have been performed and data have been collected about PD patients and COVID-19 in the first six months of the pandemic with cohorts up to 2500 patients, but only few in the following months and years. In the meantime with millions infected people worldwide, also the number of infected PD patients is supposed to be higher. This gap has to be filled to understand a possible link between PD and COVID-19..”

  1. It would be interesting to report data on patients with atypical parkinsonism (MSA, PSP, CBD). These patients usually display akinetic parkinsonism and might have similar findings compared to PD.

Response to 8: Data on patients with an atypical parkinsonism were not included, not specified or just not mentioned in the big cohort studies in Italy, France or Spain. Cilia et al., 2020 even excluded patients with APS. There is unfortunately no reliable data addressing these rare diseases. To meet your concern we added a sentence in the conclusion, that this is a limitation.

  1. Grammar and style: they are overall satisfying, but I suggest to replace “can cause” with “might” in the abstract.

Response to 9: Suggestion thankfully accepted and implemented.

Reviewer 3 Report

Authors herein present a well written overview about the relationship between SARS-CoV-2 infection and parkinson disease.

They cover almost all the emerging issues and points around this topic.

In both our out-patient clinic and neurocovid unit, we experienced so far long-lasting (up to six months) changing in motor/non motor control of few PD patients, as well as transient deterioration limited to the hyper-acute phase of Covid19. I wonder, if data were available on this point: could the Authors please comment on the duration of the motor/non motor deterioration in PD patients after Covid19 onset (at page 4, point 3 of the review)?

I recommend a careful revision of the paper to avoid few minor English mistakes (e.g. in Fig 1 legend: "...b) in PD there are to hypothetical not fully...").

Author Response

Authors herein present a well written overview about the relationship between SARS-CoV-2 infection and parkinson disease.

They cover almost all the emerging issues and points around this topic.

In both our out-patient clinic and neurocovid unit, we experienced so far long-lasting (up to six months) changing in motor/non motor control of few PD patients, as well as transient deterioration limited to the hyper-acute phase of Covid19.

  1. I wonder, if data were available on this point: could the Authors please comment on the duration of the motor/non motor deterioration in PD patients after Covid19 onset (at page 4, point 3 of the review)?

Response to 1: Dear Reviewer thanks for revising our review. We added a comment to your point on page 4, point 3. (“This study observed patients for a period of three months only, so data on long term effects, worsening or remission, were not collected.”)
A lot of data was published in the first 6 months of the pandemic, but very few after this. In the mentioned study, published August 2020, patients were observed from January till April 2020 only, so long term data were not collected.
Studies about the long-term effect (post-COVID-19/ post-acute sequelae of COVID-19 (PASC)) we discussed in point 6 of the review.

  1. I recommend a careful revision of the paper to avoid few minor English mistakes (e.g. in Fig 1 legend: "...b) in PD there are to hypothetical not fully...").

Response to 2: Thanks a lot for mentioning this typo. It should be “in PD there are TWO hypothetical not fully ….”. We corrected this.